# Inequity in access to personalized medicine in France: Evidences from analysis of geo variations in the access to molecular profiling among advanced non-small-cell lung cancer patients: Results from the IFCT Biomarkers France Study

**Samuel Kembou Nzale[1], William B. Weeks[2], L'Houcine Ouafik[3], Isabelle Rouquette[4], Michèle Beau-Faller[5], Antoinette Lemoine[6], Pierre-Paul Bringuier[7], Anne-Gaëlle Le Coroller Soriano[8]\*, Fabrice Barlesi[9], Bruno Ventelou[1]**

1 Aix-Marseille Univ., CNRS, EHESS, Centrale Marseille, AMSE, Marseille, France, 2 The Dartmouth Institute for Health Policy and Clinical Practice, Williamson Translational Building, DHMC, Lebanon, NH, United States of America, 3 Assistance Publique Hôpitaux de Marseille, Service de Transfert d'Oncologie Biologique, Aix-Marseille Univ, Marseille, France, 4 Institut Universitaire du Cancer de Toulouse, Oncopôle, Service d'Anatomie Pathologique, Toulouse, France, 5 Centre Hospitalier Universitaire de Hautepierre, Laboratoire de Biochimie et de Biologie Moléculaire & Plate-forme de Génomique des Cancers, Strasbourg, France, 6 Assistance Publique-Hôpitaux de Paris, Groupe Hospitalier des Hôpitaux Universitaires Paris-Sud, Service d'Oncogénétique- Oncomolpath, Université Paris 11, Villejuif, France, 7 Hôpital Edouard Herriot, Service d'Anatomie et de Cytologie Pathologique, Hospices Civils de Lyon, Université Claude Bernard Lyon 1, Lyon Cancer Research Center, UMR 1057 INSERM, Lyon, France, 8 Mixed Research Unit 912, Institute of Research and Development, National Institute of Health and Medical Research, Paoli Calmettes Institute, Aix-Marseille University, Marseille, France, 9 Assistance Publique Hôpitaux de Marseille, Multidisciplinary Oncology and Therapeutic Innovations Department, Aix-Marseille Univ, Centre d'Investigation Clinique, Marseille, France

\* anne-gaelle.le-coroller@inserm.fr

## Abstract

In this article, we studied geographic variation in the use of personalized genetic testing for advanced non-small cell lung cancer (NSCLC) and we evaluated the relationship between genetic testing rates and local socioeconomic and ecological variables. We used data on all advanced NSCLC patients who had a genetic test between April 2012 and April 2013 in France in the frame of the IFCT Biomarqueurs-France study (n = 15814). We computed four established measures of geographic variation of the sex-adjusted rates of genetic testing utilization at the "*département*" (the French territory is divided into 94 administrative units called '*départements*') level. We also performed a spatial regression model to determine the relationship between département-level sex-adjusted rates of genetic testing utilization and economic and ecological variables. Our results are the following: (i) Overall, 46.87% lung cancer admission patients obtained genetic testing for NSCLC; département-level utilization rates varied over 3.2-fold. Measures of geographic variation indicated a relatively high degree of geographic variation. (ii) there was a statistically significant relationship between genetic testing rates and per capita supply of general practitioners, radiotherapists and

**Data Availability Statement:** All relevant data are within the manuscript and its Supporting Information files.

**Funding:** The author(s) received no specific funding for this work.

**Competing interests:** The authors have declared that no competing interests exist.

surgeons (negative correlation for the latter); lower genetic testing rates were also associated with higher local poverty rates. French policymakers should pursue effort toward deprived areas to obtain equal access to personalized medicine for advanced NSCLC patients.

## 1. Introduction

Personalized medicine represents an opportunity to improve patients' outcomes by allowing physicians to use technological tools that determine whether patients are likely to benefit from specific treatments [1]. A potential barrier to personalized treatment relies on access to genetic testing, that must inform that treatment. In an effort to improve care outcomes, France has undertaken to make genetic testing routinely available to patients and physicians who treat them.

In 2006, the French National Cancer Institute (INCa) funded 28 regional genetic centers designed to facilitate access to molecular profiling of cancer patients [2]. Molecular profiling is particularly important for lung cancer patients because of the very high rates of genetic alterations in lung cancer, compared to other cancers [3]. In France, at least one molecular alteration was found in 43.2% of current or previous smokers' lung cancers and 74.8% of non-smokers' lung cancers and guidelines were developed to ensure routine use of molecular profiling among lung cancer patients [4]. Genetic testing for lung cancer enters the category of essential care for which difficulties in access can be detrimental to patients.

The INCa and the French Cooperative Thoracic Intergroup (IFCT) collected data from 28 regional centers to determine what kind of genetic mutations patients with advanced non-small-cell lung cancer (NSCLC) – a cancer for which molecular profiling is recommended– had and what their clinical outcomes were; they concluded that routinely nationwide profiling is feasible and offers patients a clinical benefit albeit at a 'non-negligible financial cost' [4]. However, that study did not determine whether uptake of this technology varied according to different ecological factors that might influence local use of genetic testing, such as socioeconomic status, the local supply of genetic testing centers, or the local supply of physicians. To examine these relationships, we conducted an analysis of geographic variation in the rates of the French département-level use of genetic profiling for NSCLC and explored associations between those rates and département-specific ecological variables that might explain differences in utilization rates, with an eye toward understanding inequity of access.

## 2. Materials and methods

### 2.1. Data sources, sample definitions, and variable descriptions

We used data on all advanced NSCLC patients who had a genetic test between April 2012 and April 2013 in France in the frame of the IFCT Biomarqueurs-France study. The Biomarqueurs-France study was approved by a national ethics committee for observational studies (Comité d'Evaluation des Protocoles de Recherche Observationnelle), by the French Advisory Committee on Information Processing in Material Research in the Field of Health (Comité Consultatif sur le Traitement de l'Information en Matière de Recherche dans le Domaine de la Santé), and by the National Commission of Informatics and Liberty (CNIL), according to French laws.

The Biomarqueurs-France study sought to calculate the incidence and consequences of molecular alterations among patients with advanced NSCLC [5]. To do this, between April 2012 and April 2013, the project collected data on patients diagnosed with advanced NSCLC who were referred by their physician for genetic testing; hypothetically, all advanced NSCLC patients should have been identified because genetic profiling is recommended for their evaluation during routine care.

During that period, data from 15,814 unique patients with NSCLC patients were collected [4]. Those data included a unique prescribing physician identifier that indicated the département in which the physician who ordered the genetic test worked (mainland France is divided into 94 administrative units called '*départements*'; these administrative units are the basis for the organization of most social services). In France, patients are not restricted to using healthcare services in the département in which they live. To estimate the number of tests provided to patients who lived in a given département, we assumed that patients who obtained these tests did so using the same in- and out-of-département patterns that patients who had been admitted for lung cancer did. Therefore, from Agence technique de l'information sur l'hospitalisation (ATIH) [6], we obtained data on admissions that had a primary diagnosis for lung cancer (defined as ICD 10 codes C34) [7], during the same period; these data include both the département in which the patient lived, and the département in which the patient was admitted. For each département, we determined where unique patients living in that département were admitted for lung cancer. For the entire country, we found 33,740 patients diagnosed and admitted for lung cancer of which 80% (26,900) are presumably classified as NSCLC patients. A calculation of the precise coverage rates of genetic testing should have had the population of advanced NSCLC in the denominator for all départerments, as this is the population for which the test is medically recommended during routine care. Our data does not allow to estimate these testing rates (the proportion of "advanced" at the sub-level is missing); we find nonetheless very high discrepancies in the use of tests across départements, which clearly suggests under-use for some départements in France 2012.

To reallocate healthcare utilization in the département of residence of the patient, we used the Dartmouth Atlas Project's indirect method and the département-level number of lung cancer admissions [6] of males and females aged 20–99 in 2012–2013. This method(not originally invented for the Dartmouth Atlas Project) is a classical indirect standardization that consists in correcting the epidemiological ratios measured at a given area by demographic characteristics of this area [8]. We were then able to generate sex-adjusted rates of patients who received genetic testing per 100 lung cancer admissions for each département, with département-level reallocated tests utilization in the numerator and the département-level sex-specific population of lung cancer patients in the denominator [8]. We excluded Somme (département 80) and Corsica (départements 20A, 20B) because there appeared to be an error in data collection on the number of patients who had genetic tests done there. Therefore, for 93 départements in mainland France, we used established methods to calculate 4 common measures of geographic variation in the per capita use of genetic testing: (1) the extreme ratio, (2) the interquartile ratio, (3) the coefficient of variation and the systematic component of variation (SCV) [9–11].

From ATIH [6], Institut National de la Statistique et des Etudes Economics (INSEE) [12] and Système National d'Information Inter-Régimes de l'Assurance Maladie (SNIIRAM) [13], we obtained 2 types of ecological variables that we thought might influence the use of molecular testing (see S1 Appendix for detailed information on our different data source). First, we hypothesized that the per capita département-level overall use of the healthcare system or supply of healthcare resources that might be consumed in the diagnosis and treatment of NSCLC could influence testing utilization rates. Therefore, we obtained the overall per capita hospitalization rate and the per capita number of general practitioners, surgeons, oncologists,

pathologists, and radiotherapists from national databases [13] and included them in the modelling. We included radiotherapists because their supply might be an indicator of higher technology available within a particular département. We also included dummy variables to account for the presence of a referral cancer hospital and the presence of a genetic testing center in each département. Second, because several studies found that the socio-economic status of the patient is a prominent determinant of high quality cancer care [14, 15] and type of care received by non-small cell lung cancer patients [14–19], from the same sources [12, 13] we obtained département-level measures of local economic distress: the poverty rate (a dummy was created for *départements* with poverty rates superior to 15%), and the proportion of people receiving "Couverture Maladie Universelle Complémentaire" (CMU-C), a supplemental health insurance that is only given to those whose income is below a particular level. We provide results for patients aged 18–99 and for the specific group of patients aged 60 and older. The 60 and older had the large majority of lung cancer admissions (72.2%) and genetic tests (65.9%).

We used 2 methods to determine whether these ecological variables explained geographic differences in département-level sex-adjusted per capita genetic testing utilization rates. First, we used Ordinary Least Square (OLS) regression analysis to model the relationship between sex-adjusted rates of genetic testing for NSCLC and the ecological factors that we considered. Second, we tested for spatial autocorrelation by calculating the global Moran's I statistic. Since spatial autocorrelation was evident (i.e., Moran's I = 0.28 and the associated p-value <0.001), we used a spatial error-lag regression model (weighting departmental results using a Rook criterion for the contingency matrix). See S3 Appendix for Moran's scatterplots and maps of Local Indicators of Spatial Association. A spatial auto-correlation modeling allows the correction of plausible links between the error terms of two adjacent regions. We modeled per capita use of genetic testing as the dependent variable for all patients, and we performed a sensitivity analysis using only patients aged 60 and older. For each sample, a parsimonious version of the regression is given -with 10% as a criterion for the variable's selection. We show results that account for the correction of spatial autocorrelation.

We used R © version 3.6.1 to perform all econometric analyses and GeoDa © 1.14 to perform our spatial analyses.

## 3. Results

### 3.1. Geographic variation in utilization rates of genetic testing

In mainland France, between April 2012 and April 2013, for every 100 lung cancer admissions, 46.87 patients aged 20–99 (and 42.82 patients aged 60–99) obtained genetic testing for NSCLC (Table 1). Rates of genetic testing per 100 lung cancer admissions ranged over 3-fold for both age groups: from 23.75 to 77.32 for patients aged 20–99 (and from 21.68 to 74.68 for older patients). Nièvre (département 58) had the lowest rates and Côtes-d'Amor (département 22) had the highest rates for both age groups. Extreme and inter-quartile ratios were similar for both age groups as were the coefficient of variation and systematic component of variation (which, being greater than 5, indicated a high degree of geographic variation) [20].

The Fig 1 provides a map showing quintiles of rates of use of genetic testing for NSCLC among those aged 20–99 (left) and those aged 60–99 (right). For both age groups, rates were generally lowest for department in the Champagne-Ardenne-Lorraine and Languedoc-Roussillon regions and in central France (Our computed individual rates as well as a numbered map of French départements are provided in S4 and S5 Appendices).

**Table 1. National rates of use of molecular profiling in France for advanced non-small cell lung cancer and common measures of geographic variations, April 2012 –April 2013.**

|  | Age 20–99 | Age 60–99 |
|---|---|---|
| National rate | 46.87 | 42.82 |
| Minimum rate | 23.75 | 21.68 |
| Maximum rate | 77.32 | 74.68 |
| Extreme ratio | 3.25 | 3.43 |
| Inter-quartile ratio | 1.40 | 1.44 |
| Standard deviation | 12.08 | 11.89 |
| Coefficient of variation | 0.25 | 0.27 |
| Systematic component of variation x 10 | 5.40 | 6.02 |

Rates are presented per 100 advanced non-small cell lung cancer admission aged 20–99 or 60–99

## 3.2. Results of the regression analyses

Our spatial regression models indicated that the per capita supply of surgeons, general practitioners and radiotherapists were most strongly (the former negatively so) associated with use of genetic testing (**Table 2**). We also found that neither the dummy 'living in a département with a genetic testing center' nor the dummy 'living in a department possessing a referral cancer hospital' was associated with departmental use of genetic testing. We also found that the local poverty rate was negatively associated with utilization rates: For the 20–99 population of patients, deprived departments are associated with a 10% lower proportion of use of genetic testing technologies over the period (this proportion is 8% for the 60–99). To further assess the robustness of our principal result and account for the possible collinearity between our variables capturing physicians' densities, we have estimated 5 different models with the density of general practitioner being the pivot variable (other densities are accounted for progressively from Model 1 to 5). Results are in S2 Appendix; our main observations still hold.

The usual interpretation of coefficients remains in the spatial error model: 0.11 for instance captures the slope (assuming linearity) of the rate of genetic testing to the density of general practitioners.

Apart from regression analyses, one could also use spatial analytical tools to visualize the relationships between our computed rates of genetic testing and our ecological variables [21–23]. We have used the bivariate Moran's scatterplots as well as the Local Indicators of Spatial Association (LISA) maps to better capture what we aim at depicting. Overall, these maps do not only validate our assumption of spatial correlation between our variables, but they also enable us to visualize areas where we have the most significant clusters (further maps and results are relegated to S3 Appendix). For instance, we provide below, in Fig 2, the bivariate Moran's scatterplot between our computed rates and the poverty rate. The graph confirms the negative association between poverty rates and the rates of genetic testing (already seen in our regression analyses). As displayed on the maps, for the entire sample, the Moran's I is equal to -0.082 (significant at 5%) and for the subsample of old only, it is equal to -0.11 (significant at 1%). The relationship between poverty rate and computed rates of genetic testing seems stronger for old-age groups.

The LISA map, in Fig 3, shows the most significant clusters that drive this relationship. There are 5 départements in the East where low poverty rates go together with relatively high testing rates. The 4 départements with high poverty rates and low testing rates are however not as grouped. The maps are visually similar for the entire sample and for the older-age group.

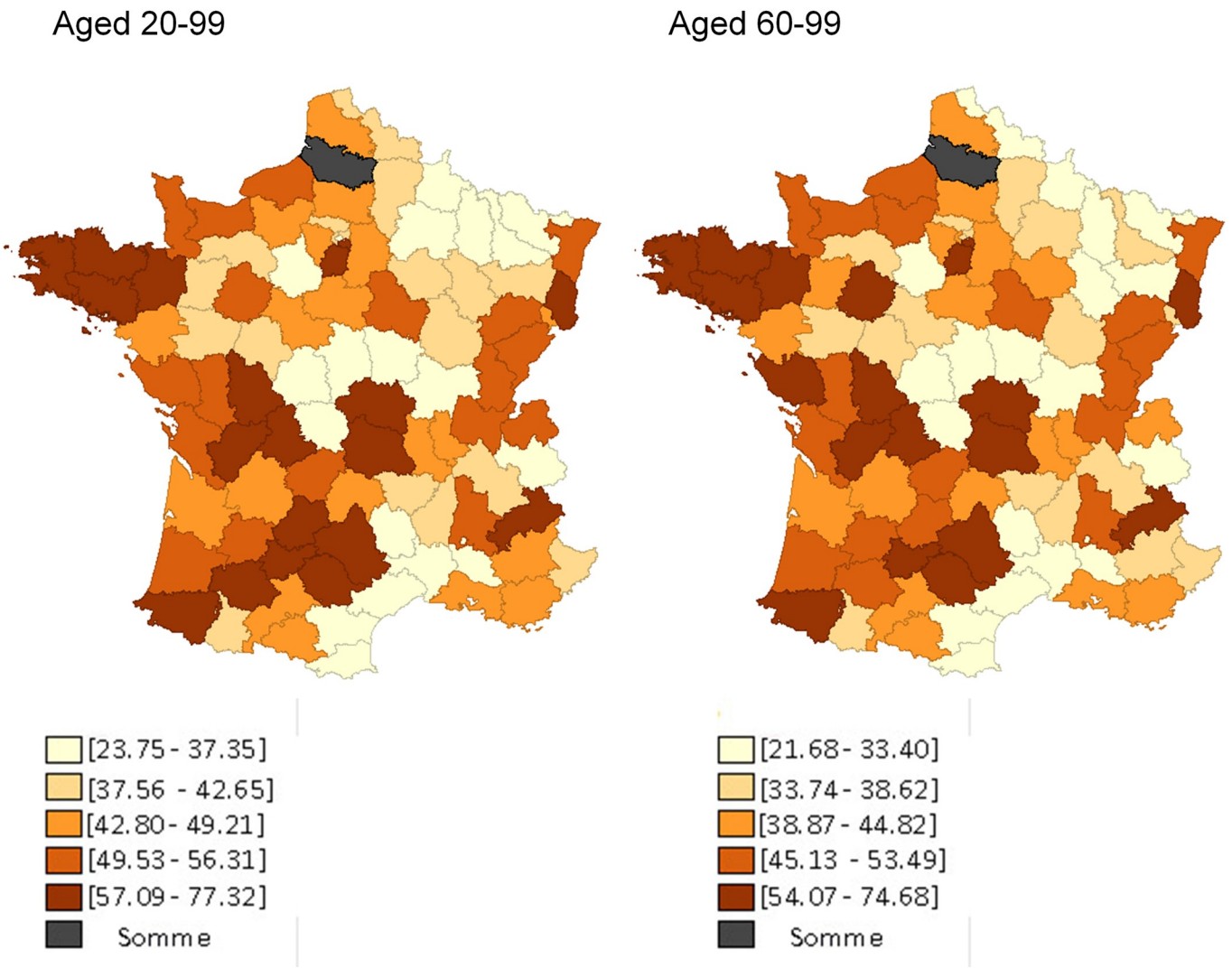

**Fig 1. Département-level quintiles of rates of genetic testing for NSCLC in France among inhabitants aged 20–99 (left) and those aged 60–99 (right), April 2012 –April 2013.** For each département, we know where unique patients living in that département were admitted for lung cancer. Using that information, we calculated the département-specific proportion of hospital stays (for males and females, separately) that were provided to patients who lived in that département and in any other département. For instance, during the study's period, among males, there were 68 lung cancer admissions in Loir-et-Cher (department 41): 96% of those admissions were for patients who lived in Loir-et-Cher, but 2.5% were for patients who lived in Indre-et-Loire (department 36) and 1.5% were for patients who lived in Loiret (department 45). To estimate the number of genetic tests done on patients who lived in a particular département, we then allocated tests obtained in a département according to how patients had been admitted for lung cancer. Therefore, continuing our example, we allocated the 30 genetic tests that were ordered on males by physicians working in Loir-et-Cher accordingly: 28.78 (96%) to Loir-et-Cher, 0.77 (2.5%) to Indre-et-Loire, and 0.44 (1.5%) to Loiret. We then added all allocated tests expected to have been received by males and females, separately, who lived in each département. Data from the départements 'Somme' and 'Corsica (North and South)' are missing.

## 4. Discussion

We studied geographic variation in rates of use of personalized medicine for advanced NSCLC in France. One value-added of this work is to bring together different sources of data to demonstrate and explain geographical differences in use of genetic testing. We found substantial variations across *départements* and several correlates with ecological variables. Rates of use of personalized medicine technologies were affected by the supply of health professionals as well as the deprivation of the living area of the patient. We were initially surprised to discover an inverse relationship between the per capita supply of surgeons and the use of genetic testing;

**Table 2. Multivariate analysis of geographic variation of molecular profiling use in France for advanced non-small cell lung cancer.** April 2012 –April 2013.

| Ages of population included and models | Spatial regression models | | | |
|---|---|---|---|---|
| | 20–99 | 20–99 (parsimonious model) | 60–99 | 60–99 (parsimonious model) |
| Poverty rate (*dummy w. ref = rate >15%*) | -7.54** (3.68) | -9.91*** (3.09) | -6.86* (3.58) | -8.64*** (3.03) |
| Per capita supply of | | | | |
| General practitioner (per 100,000) | 0.11** (0.05) | 0.08** (0.04) | 0.11** (0.05) | 0.08** (0.04) |
| Surgeons (per 100,000) | -1.75* (1.01) | -1.96** (0.91) | -1.84* (0.98) | -2.24** (0.93) |
| Radiotherapists (per 100,000) | 6.47* (3.93) | 6.59* (3.65) | 7.75** (3.82) | 8.12** (3.55) |
| Pathologists (per 100,000) | -3.40 (2.48) | | -3.23 (2.41) | |
| Oncologists (per 100,000) | 0.95 (4.08) | | 0.77 (3.97) | |
| Beds (per 100,000) | 0.10 (0.11) | | 0.19 (0.10) | |
| Per-capita admission rate (per 100,000) | -1.13 (0.90) | | -1.56* (0.87) | -0.80 (0.65) |
| Presence of a genetic testing center (dummy) | 2.20 (4.14) | | 1.52 (4.03) | |
| Presence of a referral cancer hospital (dummy) | -2.35 (4.19) | | -2.40 (4.08) | |
| Proportion receiving CMUC (per 100,000) | -0.48 (0.49) | | -0.38 (0.49) | |
| Constant | 60.97*** (14.95) | 45.19*** (2.85) | 62.16*** (14.52) | 54.24*** (11.08) |
| *Observations* | 93 | 93 | 93 | 93 |
| *Log Likelihood* | -346.38 | -348.23 | -34.71 | -345.27 |
| *sigma²* | 96.11 | 100.36 | 90.77 | 94.38 |
| *Akaike Inf. Crit.* | 720.77 | 710.46 | 715.42 | 706.53 |
| *Wald Test (df = 1)* | 13.07*** | 11.78*** | 12.90*** | 10.94*** |
| *LR Test (df = 1)* | 7.41*** | 8.14*** | 7.32*** | 7.21*** |

All coefficients (and standard errors) are shown.

*p<0.1

**p<0.05

***p<0.01

however, it is possible that surgeons influence the therapeutic choice in favor of a rapid surgical intervention and then use genetic testing less frequently. A higher per capita supply of radiotherapists was perhaps reflecting a greater overall supply of advanced cancer healthcare services in the local setting. However, the fact that the presence of a genetic testing center or a referral cancer hospital in the département was not a statistically significant predictor of genetic testing rates provides an interesting result. It actually tends to validate the territorial

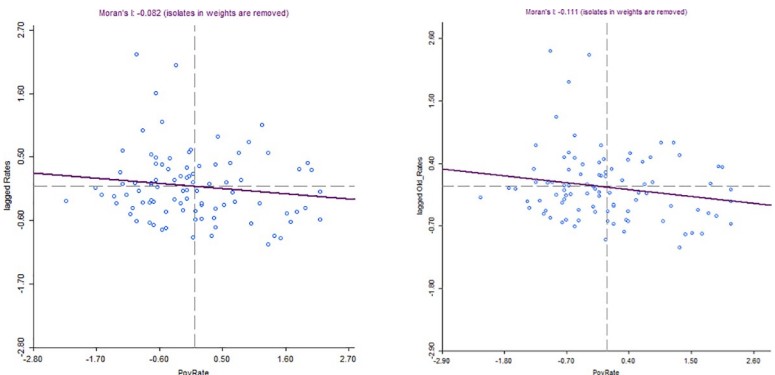

**Fig 2. Bivariate Moran scatterplots between poverty rate and genetic testing rates for NSCLC in France among inhabitants aged 20–99 (left) and those aged 60–99 (right), April 2012 –April 2013.**

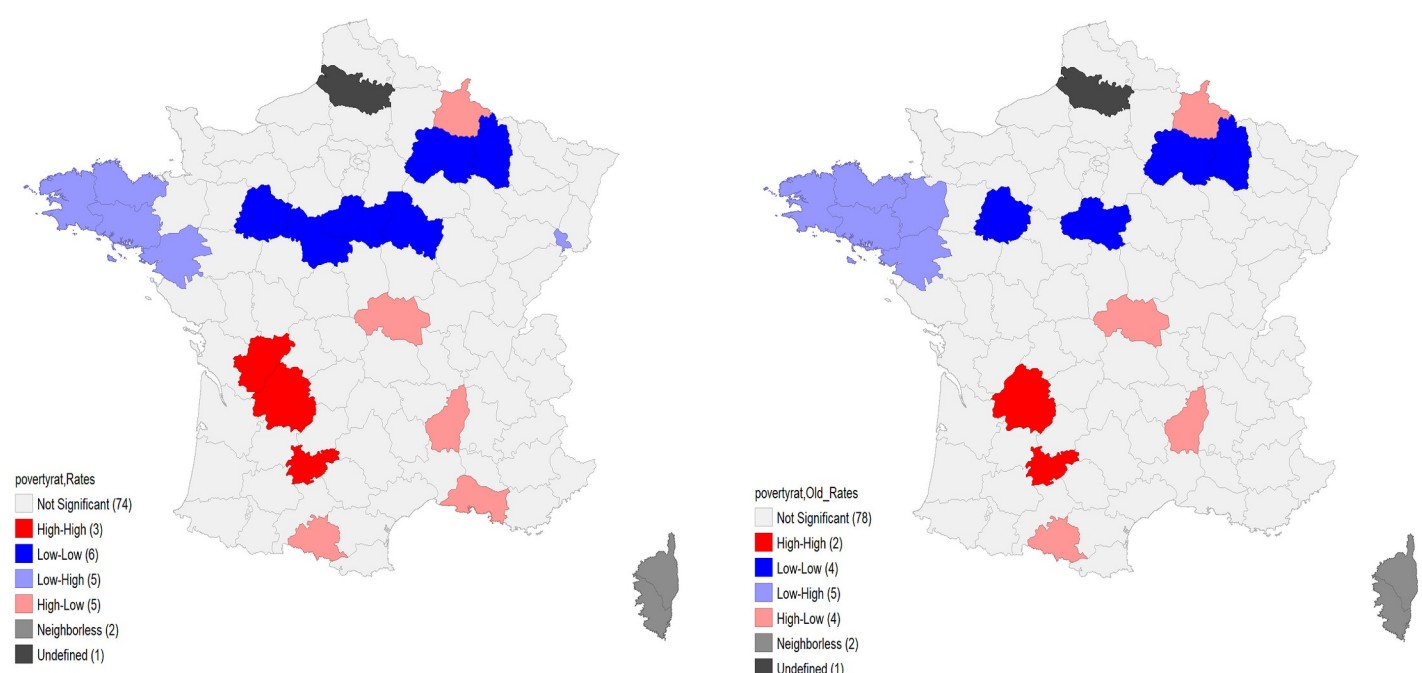

**Fig 3.** Bivariate LISA maps between poverty rate and genetic testing rates for NSCLC in France among inhabitants aged 20–99 (left) and those aged 60–99 (right), April 2012 –April 2013.

grid of the genetic centers and reference cancer hospitals across France and their effective communication with the decentralized hospitals.

We also found that patients living in high-poverty *départements* were less likely to receive genetic testing after correcting for other explanatory factors. This inequality of access observed is an issue for the French healthcare system which claims to provide free and equitable access to care for all cancer patients. There are recent US studies that have documented the link between NSCLC patients' place of residence and their access to treatments: [17, 19]. Yorio et al. [17] have shown in a study done within a single academic medical center in Texas that socioeconomically disadvantaged patients with stage I-III NSCLC were less likely to receive 'standard' therapy; while Jiang et al. [17] showed that Nebraska NSCLC patients residing in high poverty neighborhoods were twice less likely to receive surgery than those in low poverty neighborhoods. In our study, we complement earlier work by giving evidence that access to personalized medicine for NSCLC patients is influenced by the social gradient of the department in which the patient lives. Although French authorities determined that routinely nation-wide genetic profiling is feasible, our findings suggest that it is currently inequitable and that a focus on *départements* with high poverty levels would reduce that inequity.

Our analysis has several limitations. First, through the reallocation process, we used administrative data for lung cancer admissions from 2012–2013 to estimate where patients who obtained genetic testing lived. Patients might use different healthcare utilization patterns for genetic testing and hospitalization for lung cancer, and future studies should collect data on patients' residence to more accurately evaluate their access to genetic testing. Second, we were not able to observe the precise proportion of advanced non-small cell lung cancer among the total lung cancer in each *département*, which would be a better denominator for utilization rates. We believe however that the expected differences across *départements* in this proportion cannot explain such high variations in utilization rates (anyway, in the literature we are not

aware of any proven relationship between poverty rate and the proportion of NSCLC). Finally, use of genetic testing for advanced NSCLC in 2012–2013 might not reflect current utilization patterns; there is hope that the equality of access has improved in recent years [24].

## 5. Conclusion

Our study suggests that *départemental* economic distress might negatively impact routine use of genetic testing. On the supply side, potential reasons for lower rates in certain *départements* can be the fact that, it is time-demanding for prescribing physicians to require these tests (both administrative and care coordination costs). Moreover, not all the genetic platforms are equipped to provide all the tests, probably limiting a-priori physicians' decisions to require a genetic test. Future research should explore reasons for this low-access and seek to better explain variations in rates that we found. What we, however, consider as a key policy recommendation for this study is that French policymakers should target deprived areas to provide equal access to personalized medicine for advanced NSCLC patients. Another lever of public policy could be to reinforce territorial access to specialized health workforce which implies addressing the challenges of attractiveness and retention in French underserved areas.

## Supporting information

**S1 Appendix. Summary of our data source.**
(DOCX)

**S2 Appendix. Spatial regression models to test the stability of our results.** Significance levels are the same as reported in Table 2 above.
(DOCX)

**S3 Appendix. Moran scatterplots and LISA maps.**
(DOCX)

**S4 Appendix. Numbered map of French départements.**
(DOCX)

**S5 Appendix.** Rates of genetic testing rates for NSCLC in France among inhabitants aged 20–99 (left) and those aged 60–99 (right), April 2012 –April 2013.
(DOCX)

**S1 Data.**
(XLSX)

## Author Contributions

**Conceptualization:** Anne-Gaëlle Le Coroller Soriano, Fabrice Barlesi, Bruno Ventelou.

**Data curation:** Samuel Kembou Nzale.

**Formal analysis:** Samuel Kembou Nzale, William B. Weeks.

**Investigation:** L'Houcine Ouafik, Isabelle Rouquette, Michèle Beau-Faller, Antoinette Lemoine, Pierre-Paul Bringuier, Fabrice Barlesi.

**Methodology:** Samuel Kembou Nzale, William B. Weeks, Anne-Gaëlle Le Coroller Soriano, Bruno Ventelou.

**Project administration:** Samuel Kembou Nzale, Anne-Gaëlle Le Coroller Soriano, Fabrice Barlesi.

**Supervision:** William B. Weeks, Anne-Gaëlle Le Coroller Soriano, Fabrice Barlesi, Bruno Ventelou.

**Validation:** L'Houcine Ouafik, Isabelle Rouquette, Michèle Beau-Faller, Antoinette Lemoine, Pierre-Paul Bringuier, Fabrice Barlesi.

**Writing – original draft:** Samuel Kembou Nzale, William B. Weeks, Anne-Gaëlle Le Coroller Soriano, Bruno Ventelou.

**Writing – review & editing:** L'Houcine Ouafik, Isabelle Rouquette, Michèle Beau-Faller, Antoinette Lemoine, Pierre-Paul Bringuier, Anne-Gaëlle Le Coroller Soriano, Fabrice Barlesi.

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
