## [Decision Letter · Decision Letter 0]

7 Nov 2019

PONE-D-19-22289

Inequity in access to personalized medicine in France: Evidences from analysis of small area variations in the access to molecular profiling among advanced non-small-cell lung cancer patients: Results from the IFCT Biomarkers France Study.

PLOS ONE

Dear Mrs Le Corroller Soriano,

Thank you for submitting your manuscript to PLOS ONE. After careful consideration, we feel that it has merit but does not fully meet PLOS ONE’s publication criteria as it currently stands. Therefore, we invite you to submit a revised version (minor revision) of the manuscript that addresses the points raised during the review process.

We would appreciate receiving your revised manuscript by Dec 22 2019 11:59PM. To enhance the reproducibility of your results, we recommend that if applicable you deposit your laboratory protocols in protocols.io, where a protocol can be assigned its own identifier (DOI) such that it can be cited independently in the future. For instructions see: http://journals.plos.org/plosone/s/submission-guidelines#loc-laboratory-protocols

We look forward to receiving your revised manuscript.

Kind regards,

Alberto d'Onofrio, Ph.D.

Academic Editor

PLOS ONE

Journal Requirements:

Additional Editor Comments:

Dear Authors,

your manuscript has been reviewed by two expert in the section and both suggested minor revisions.

Both included a quite list of changes to be included, so I recommend you to very carefully implement all of them.

A small suggestion: usually one refer to "R" as "CRAN R".

Reviewers' comments:

Reviewer's Responses to Questions

**Comments to the Author**

1. Is the manuscript technically sound, and do the data support the conclusions?

Reviewer #1: Yes

Reviewer #2: Yes

2. Has the statistical analysis been performed appropriately and rigorously? 

Reviewer #1: Yes

Reviewer #2: Yes

3. Have the authors made all data underlying the findings in their manuscript fully available?

Reviewer #1: No

Reviewer #2: No

4. Is the manuscript presented in an intelligible fashion and written in standard English?

Reviewer #1: Yes

Reviewer #2: Yes

5. Review Comments to the Author

Reviewer #1: Review of manuscript PONE-D-19-22289

The manuscript reports on a study conducted in France assessing the access to gene testing for NSCLC lung cancers. It uses geographic analysis and account for availability of health workforce, poverty in the region.

The study is overall well conducted and show clear results on the impact of poverty and on lack of access to health workforce as factors explaining the lower access to genetic testing. The use of a large sample of patients, ATIH database (which contain a lot of very under-utilized data for research), and appropriate method is really appreciated.

General comments:

The text is unclear on whether NSCLC patients included in the study correspond to all patients in France, or only a subset seen in the 28 regional cancer centers. From a quick and dirty check, with around 30000 lung cancer diagnosed per year in France, and with 80% of them being NSCLC, the present study conducted during a year would have captured 70% of all NSCLC in France. This pose the question of transferability of the results and the potential risk of bias. Who are the remaining 30% of NSCLC? They probably live in more remote area, maybe covered by large hospital although not specialized cancer center. And lack of access to a cancer center in the first place could also be associated with poverty, etc… So the present study could have been conducted in a particular subset. If these are indeed 70% of all cancers, the problem is relatively small but still worth mentioning to the reader. If less than 70% it could require further investigation.

The analysis could also include as parameter the availability of other hospital structure who could provide follow-up of NSCLC.

The source of data are not detailed enough. For health workforce density, mentioning SNIIRAM data does not allow to define if the study refers to active workforce, professionally active (active but not always providing clinical care, such as teaching), or licensed to practice. There is also a lack of information on the year of the data used, are these the density for 2019, 2018? For other indicators such as poverty rate etc, the methods are not described, there are several metrics existing to measure poverty, and the source are not clearly identifiable.

The modelling reported on table 2 includes the density of several health care occupations. These are very often highly correlated, and the modelling shows surprising results with some coefficient being positive and other being negative. It would be advisable to check if this is due to collinearity or not. It would be helpful to provide (in an annex) the model with only GP, and then reassure the reader that the progressive addition of other health care workers did not created this phenomenon of coefficient being positive and negative.

The second limitation (page 11, lines 238-242) suggest in a long parenthesis (remove parenthesis) that it is not an issue, while this could be associated with poverty and as a result with the percentage of genetic testing.

References are often misused (for example page 3, line 65, ref 5), or with wrong numbering (for example: page 4, line 101, ref 7). References should be double checked and harmonized.

Minor comments:

The title of the manuscript mentions small area variation. As the unit of analysis are French regions (departments) and include half a million inhabitants for several of them, it does not match with a label of small area.

Dartmouth Atlas Project is cited for a method of indirect standardization, as this method is not originally invented for the Dartmouth Atlas Project, but a very classical indirect standardization available in all epidemiology textbook, another reference could be used (if need be). The only addition on the indirect standardization for the reader would be to explain in two sentences how it is computed and the notion that no population data are required.

The method describe the use of a spatial error-lag regression model, this is appropriate but needs a short description of the reason to choose this method for the neophyte.

Add a reference for R software, at least which version was used.

Table 1 is unclear: what seems to be reported is the percentage of patients receive genetic testing, while it states “rate of geographical variation”. Same in the footnote, the mention of a rate per 100 advanced NSCLC, again this does not seem to be a rate.

Table 2, define the metric used (mostly these are estimates from the regression), please also provide the metrics for each variable to enable the interpretation of the results. For example, an estimate of 0.11 for general practitioner corresponds to what? Is it an increase of 0.11% (or 11%) of the percentage of genetic testing for each additional GP per XXX(how many) population?

In the conclusion line 253-255 page 11, in addition to targeting deprived areas, it would be useful as policy option to recommend reinforcing access to health workforce which implies addressing the challenges of attractiveness and retention in rural/remote areas.

Reviewer #2: Dear Editors,

The manuscript by Kembou Nzale et al is well written and it is on a very important topic. Ir provides useful inferences and suggestions,which both can help to improve Public Health in France and, if the study s extended to other countries, elsewhere.

I suggest minor revisions since the geostatistical analysis of this work could be improved to provide further insights.

List of points to be amended:

1. Authors claim that the study was conducted in “mainland France” which “is divided into 94 94 administrative units called ‘départements”. However, as also shows by the two maps, their study did not include Corse Island, apparently. Authors must clarify this point.

2. The authors employed the Moran Index to assess the extent of spatial autocorrelation of the maps. It would be of interest to insert the Moran scatterplots See: Robertson, Chris, Chiara Mazzetta, and Alberto D’Onofrio. "Regional variation and spatial correlation." Chapter 5 in P. Boyle and M. Smans (Eds.), Atlas of Cancer Mortality in the European Union and the European Economic Area 1993-1997 (2008): 91-113; Anselin L (1995) Local indicators of spatial association Geographical Analysis 27 93–115

3. Authors wrote: “autocorrelation was evident (i.e., Moran’s I was <0.001” This is a misprint:, probably the authors refers to the Z--value associated to the Moran I. They mus provide the value of the Moran I.

4. The authors provide two maps for two groups of age classes. The maps are visually similar, i.e. they seem highly spatially correlated. However it would be appropriate to quantify the degree of similarity between them by means of the “two-maps Moran’s scatterplot” and the bivariate Moran’s index, which were both defined in the above mentioned paper by Ronertson et al. See also : d’Onofrio, Alberto, et al. "Maps and atlases of cancer mortality: a review of a useful tool to trigger new questions." ecancermedicalscience 10 (2016).

5. The authors used some spatial data of “ecological” variables: it would be of interest for the reader if the related data would be available. Equally, it would be of extreme interest as an assessment with the univariate Moran index, and (even more interesting) to compare by means of the bivariate Moran index each of these maps with the map of the “rates of use of genetic testing for NSCLC” in the class age 1-59 (and for that in the class 60-99).

6. A practical suggestion: authors ought to include a numbered map of French department, in order that the non French reader can localize the mentioned departments.

6. PLOS authors have the option to publish the peer review history of their article (what does this mean?). If published, this will include your full peer review and any attached files.

Reviewer #1: Yes: Dr Mathieu Boniol

Reviewer #2: No

---

## [Author Response · Author response to Decision Letter 0]

25 May 2020

All relevant data are now within the manuscript's supporting Information files. Responses to reviewers ares in the file Responses to reviewer.

---

## [Decision Letter · Decision Letter 1]

27 May 2020

Inequity in access to personalized medicine in France: Evidences from analysis of geo variations in the access to molecular profiling among advanced non-small-cell lung cancer patients: Results from the IFCT Biomarkers France Study.

PONE-D-19-22289R1

Dear Dr. Le Corroller Soriano,

We are pleased to inform you that your manuscript has been judged scientifically suitable for publication and will be formally accepted for publication once it complies with all outstanding technical requirements.

With kind regards,

Alberto d'Onofrio, Ph.D.

Academic Editor

PLOS ONE

Additional Editor Comments (optional):

Reviewers' comments:

Reviewer's Responses to Questions

**Comments to the Author**

1. If the authors have adequately addressed your comments raised in a previous round of review and you feel that this manuscript is now acceptable for publication, you may indicate that here to bypass the “Comments to the Author” section, enter your conflict of interest statement in the “Confidential to Editor” section, and submit your "Accept" recommendation.

Reviewer #1: All comments have been addressed

Reviewer #2: (No Response)

2. Is the manuscript technically sound, and do the data support the conclusions?

Reviewer #1: Yes

Reviewer #2: Yes

3. Has the statistical analysis been performed appropriately and rigorously? 

Reviewer #1: Yes

Reviewer #2: Yes

4. Have the authors made all data underlying the findings in their manuscript fully available?

Reviewer #1: No

Reviewer #2: Yes

5. Is the manuscript presented in an intelligible fashion and written in standard English?

Reviewer #1: Yes

Reviewer #2: Yes

6. Review Comments to the Author

Reviewer #1: All comments have been addressed and the authors should be congratulated for this revision which improved an already good and useful analysis.

Reviewer #2: Dear Authors,

you have done an excellent work of revision and your manuscript can now be accepted as it is.

Kind Regards,

An Anonymous Referee

7. PLOS authors have the option to publish the peer review history of their article (what does this mean?). If published, this will include your full peer review and any attached files.

Reviewer #1: Yes: Mathieu Boniol

Reviewer #2: No

---

## [Editor Report · Acceptance letter]

9 Jun 2020

PONE-D-19-22289R1 

Inequity in access to personalized medicine in France: Evidences from analysis of geo variations in the access to molecular profiling among advanced non-small-cell lung cancer patients: Results from the IFCT Biomarkers France Study. 

Dear Dr. Le Corroller Soriano:

I'm pleased to inform you that your manuscript has been deemed suitable for publication in PLOS ONE. Congratulations! Your manuscript is now with our production department. 

Kind regards, 

on behalf of

Dr. Alberto d'Onofrio 

Academic Editor

PLOS ONE